# Assessing Medical Students’ Confidence towards Provision of Palliative Care: A Cross-Sectional Study

**DOI:** 10.3390/ijerph18158071

**Published:** 2021-07-30

**Authors:** Shih-Ya Leung, Eliza Lai-Yi Wong

**Affiliations:** JC School of Public Health and Primary Care, Faculty of Medicine, The Chinese University of Hong Kong, Hong Kong, China; 1155108309@link.cuhk.edu.hk

**Keywords:** palliative care, confidence, medical students, end-of-life care, KAP

## Abstract

Under a surging demand for palliative care, medical students generally still show a lack of confidence in the provision in abroad studies. This cross-sectional study aims to investigate the confidence and its association with knowledge, attitude and exposure on providing palliative care among medical undergraduates with a self-administered questionnaire to improve the international phenomenon. Full-time local medical undergraduates were recruited to obtain information regarding the demographics, confidence, knowledge, attitude and exposure on palliative care; the information was collected from July 2020 to October 2020. Questions on confidence (10-items), knowledge (20-items), attitude (10-items) and exposure were referenced from validated indexes and designed from literature review. Confidence level was categorized into “Confident” and “Non-confident” as suggested by studies to facilitate data analysis and comparison. Of the 303 participants, 59.4% were “Non-confident” (95% C.I.: 53.8% to 65.0%) in providing palliative care on average. Among medical students, knowledge (*p* = 0.010) and attitude (*p* = 0.003) are significantly positively associated with the confidence to provide palliative care, while exposure to death of family/friends (*p* = 0.024) is negatively associated. This study begins an investigation on the research area in Hong Kong primarily. The confidence of local medical students should be enhanced to provide palliative care in their future. It thus highlights the importance of the medical curriculum and provides insights to remove barriers responsively to improve the overall confidence and the quality of palliative care.

## 1. Introduction

Palliative care which encompasses end-of-life care, refers to “the approach that improves the quality of life of patients and their families facing the problem associated with life-threatening illness, through the prevention and relief of suffering by means of early identification and impeccable assessment and treatment of pain and other problems, physical, psychosocial and spiritual” according to WHO’s definition [1]. Under the rapid aging population in Hong Kong, there will be a surging amount of elderly in the society [2,3]. With a higher prevalence of chronic diseases, the community has a higher demand and expectations towards the availability and quality of palliative care [2,3]. Compared to places like Australia and Taiwan, the number of palliative care physicians per 1 million population reaches 13.2 and 24.4, while the proportion is only 6.9 in Hong Kong locally [4]. There will be a foreseeable burden on clinical workers, as well as the medical system to handle the surging demands.

We analyzed the confidence or attitude of medical students to recursively assess the adequacy of training and education of palliative care [5]. Provision of end-of-life care requires both advanced skills and appropriate attitudes. A mere assessment of knowledge and teaching hours would not be able to reflect the confidence, the state of mental preparedness which includes the knowledge and willingness of the medical student [6]. A variety of factors are documented to be associated with the confidence to provide palliative care. For instance, social-cultural factors (e.g., considerations on filial piety—palliative care is violating filial piety to seniors) or economic factors (e.g., inadequate financial support—there is no revenue to support the palliative care development) [7,8,9] were identified. With medical undergraduates in progress to develop the preparedness of palliative care, research shows that medical curriculum could be effective in enhancing a higher readiness to enter the future clinical career in terms of various aspects [10,11]. A refined academic understanding over clinical knowledge, skills and techniques to handle various clinical palliative situations allows medical students to better prepare themselves in handling of end-of-life dilemmas [12]. However, the comprehensive teaching could be a double-edged sword which allows students to recognize their limitations in facing deaths of patients or in handling complex clinical situations. Thus, this might reduce their confidence in performing palliative care in the future [13]. Through the moral reasoning embedded into the courses, students could foster a positive attitude towards different palliative care issues [12]. Benefits to their respective traits through provision of palliative care could always serves as enablers to build up their corresponding confidence—palliative care could help patients having a good death or provide an emotional support before their last stage of life [12]. On the contrary, perceived barriers such as the fear to confront death of patients, or subjective personal interpretation of palliative care equivalents to giving up on patients could lead students with a reduced confidence to provide palliative care [14,15].

Throughout the current local medical curriculum, ward-based clinical training allows students with hands-on experience to in close contact with patients in handling various situation to foreshadow potential clinical dilemmas. For instance, in Hong Kong medical institutions, a total of over 24 hours of mandatory ethical training are implemented into the 6-year-curriculum in formats of lectures, case studies, facilitated discussion groups, etc. on different end-of-life topics to improve a better handling of future palliative dilemmas [16,17]. Studies show that students usually tend to have a better confidence with an increased clinical exposure [18,19,20,21,22]. On the other hand, with a higher voluntary exposure and fostered compassion to provide assistance to the vulnerable, it increases the willingness and confidence of students to provide palliative care to patients as a compassionate doctor [18,19,20,21,22]. However, research also shows that death of family/friends could have an opposite effect on the confidence due to the painful memories when facing the death of patients [23,24,25].

Compared to other places like Australia (2nd), Taiwan (6th), Germany (7th) and Netherlands (8th) with comprehensive medical systems, these countries tend to have a high ranking of palliative care in terms of the Quality of Death Index [26] along with their high resources in palliative care system while Hong Kong ranked the 22nd. From the statistics, it proposed that a higher availability of palliative care physicians could relate to the quality of palliative services [22,27,28,29,30]. However, their respective studies still show a lack of confidence and preparedness of medical students in performing palliative care. It reveals a need to assess the confidence in a highly educated setting—Hong Kong needs to identify barriers responsively to refine the palliative care services. The study aims to investigate the confidence level and the association between the confidence and the knowledge, attitude and experiential exposure of medical students using a highly educated setting—Hong Kong. The research is expected to provide responsive insights to tackle the identified barriers of medical students to provide palliative care before starting their career. In addition, the study may contribute a more comprehensive understanding in increasing the overall confidence of medical undergraduates and refining the availability and sustainability of palliative care services internationally.

## 2. Materials and Methods

### 2.1. Overview of Design

A cross-sectional survey with a structured questionnaire was adopted to investigate medical students’ confidence to provide palliative care and its association with their knowledge, attitude and exposure variables. Anonymous, self-administered questionnaires were distributed to all medical students studying in either The Chinese University of Hong Kong (CUHK) or the University of Hong Kong (HKU). In Hong Kong, The Chinese University of Hong Kong (CUHK) (Bachelor of Medicine and Bachelor of Surgery [MBChB]) and the University of Hong Kong (HKU) (Bachelor of Medicine and Bachelor of Surgery [MBBS]) are the only two institutions providing undergraduates medical bachelors, with a total of 530 students every year [31,32]. At both universities’ students undergo a Western medical education that provides a six-year curriculum, in which the first three years are pre-clinical years, followed by another three years of clinical years with increased ward exposure [17,33]. Ethics approval was obtained from the Survey and Behavioral Research Ethics of The Chinese University of Hong Kong. Informed written consent was properly embedded into the questionnaire and received to protect the rights of all respondents.

### 2.2. Study Subjects

The target population of the study is all medical students in a six-year medical curriculum in Hong Kong. Full-time CUHK and HKU medical undergraduates from Year 1 to Year 6 were included. Eligible participants were recruited through convenience sampling with a link of an anonymous, self-administered quantitative questionnaire (i.e., Google Form) sent to medical students for four months. Due to the unavailability of local data on the confidence rate of palliative care among medical undergraduates, the data from South Korea with a rate of 85.6% are referenced [34], in which, South Korea has a very similar context as Hong Kong, having a similarly advanced medical system yet a relatively low quality of palliative care (in terms of the Quality of Death Index ranking) [26]. Moreover, both medical curriculum in South Korea and Hong Kong are undergoing a Western medical education, following a six-year training system before starting their clinical career [35]. The minimal sample size to be collected for this study would be 296 with 95% of confidence level and desired relative precision level of 0.04.
(1)$$n=\frac{1.96^{2}\;x\;0.856\left(1-0.856\right)}{0.04^{2}}=296\;\left(round\;up\;to\;nearest\;1\right)$$

### 2.3. Measurements

The questionnaire was developed from the Cognitive Theory of Planned Behavior. [36] and it was composed of 56 questions under five sections: demographics, an outcome variable as the confidence to provide palliative care, and three main predictor variables, namely knowledge, attitude and exposure of palliative care referenced from abroad studies with similar contexts and literature review. It took 5–8 mins to complete.

#### 2.3.1. Socio-Demographics

Personal information regarding respondents (i.e., sex, age, year of study, occupational status of breadwinner in the household, religion, affiliated university) were collected. Occupational status of the breadwinner includes choices of white-collar (clerical, administrative and professionals), blue-collar (industrial, service and agricultural workers), or housewife/retired/unemployed. This question served as a proxy question to classify the family income-level of the participant since previous local study revealed the influence of students’ career choice with respect to the family’s income-level that follows the classification of government census [37,38]. Respondents were also categorized into those without and with religious beliefs. In abroad studies, sex and age are considered as confounders for the association between confidence and knowledge, attitude and exposure [39].

#### 2.3.2. Confidence to Provide Palliative Care

The confidence of students on palliative care was explored with modified validated tool (10-items) from Germany [33] regarding their self-estimation to carry out palliative care with a five-point Likert scale from “Very Unconfident (1)” to “Very Confident (5)”. The total score out of 50 was summed up from the 10 items, where a higher score indicates a higher confidence. As suggested from abroad studies, Likert scale 1 to 3 is classified as “Non-confidence” while Likert scale 4 to 5 is “Confidence” to facilitate data analysis and comparison [27,28,29].

#### 2.3.3. Knowledge in Palliative Care

The Palliative Care Knowledge Test (PCKT-20 items) (Cronbach’s α = 0.81) was adopted to identify the comprehensive understanding towards palliative care among medical students, which includes questions on philosophy, principles and practical knowledge (i.e., philosophy, pain, dyspnea, psychosocial and gastro-intestinal problems) with a true/false/unsure answer choice [40]. A correct answer counted as 1 mark and an incorrect answered or an unsure answer was counted as 0 mark. The total score of 20 items under five aspects is an equally weighted sum of these 20 marks: (1) Palliative care should only be provided for patients who have no curative treatment available. (2) Palliative care should not be provided along with anti-cancer treatments. (3) One of the goals of pain management is to get a good night’s sleep. (4) When opioids are taken on a regular basis, non-steroidal anti-inflammatory drugs should not be used. (5) When opioids are taken on a regular basis, non-steroidal anti-inflammatory drugs should not be used. (6) The effect of opioids should decrease when pentazocine (Talwin) or buprenorphine hydrochloride (Buprenex) is used together after opioids are used. (7) Long-term use of opioids can often induce addiction. (8) Use of opioids does not influence survival time. (9) Morphine should be used to relieve dyspnea in cancer patients. (10) When opioids are taken on a regular basis, respiratory depression will be common. (11) Oxygen saturation levels are correlated with dyspnea. (12) Anticholinergic drugs or scopolamine hydrobromide (Transderm-V) are effective for alleviating bronchial secretions of dying patients. (13) During the last days of life, drowsiness associated with electrolyte imbalance should decrease patient discomfort. (14) Benzodiazepines should be effective for controlling delirium. (15) Some dying patients will require continuous sedation to alleviate suffering. (16) Morphine is often a cause of delirium in terminally ill cancer patients. (17) At terminal stages of cancer, higher calorie intake is needed compared to early stages. (18) There is no route except central venous for patients unable to maintain a peripheral intravenous route. (19) Steroids should improve appetite among patients with advanced cancer. (20) Intravenous infusion will not be effective for alleviating dry mouth in dying patients.

#### 2.3.4. Attitude towards Palliative Care

The section composes of 10 items under two parts—perceived beliefs and barriers to assess the attitude of students towards palliative care. Perceived benefits included four items: (1) Providing palliative care enables me to promote life quality of the patients. (2) Providing palliative care can keep patient’s dignity by enabling the patient to die peacefully and having a comfortable and good death. (3) Palliative care can relieve pain and other symptoms of the patients. (4) Palliative care can provide emotional support, care and companionship for the patients. Perceived benefits will be investigated in terms of a five-point Likert scale from “Strongly Disagree (1)” to “Strongly Agree (5)” [12]. Perceived barriers included six items: (1) Providing palliative care feels like abandoning/giving up on patients. (2) Palliative care can make patients feel hopeless. (3) Palliative care can shorten patient’s life. (4) My personal inability to face dying process and distress. (5) My personal unwillingness to face dying process and distress. (6) Providing palliative care can make me feel weak and worthless about life. Perceived barriers will be investigated in terms of a five-point Likert scale from “Strongly Disagree (5)” to “Strongly Agree (1)” [12]. The overall attitude score is calculated by a sum of the 10 items to estimate the tendency of attitude of respondents by a total score of 50 [12].

#### 2.3.5. Exposure in Performing Palliative Care

Based on the literature review, five items under two categories were included as an encounter related to palliative care [4]. The exposure could be classified into (1) experiencing the death of a family member/friend or (2) clinical exposure, such as ward exposure: no. of times of discussing end-of-life issues with patients or their families; or during clinical attachment: no. of times of witnessing or following-up any end-of-life patients or their families, voluntary exposure (i.e., no. of weeks of outreach voluntary services for terminally ill patients) were included as a response in this section.

### 2.4. Data Analysis

Data was analyzed by using SPSS Statistics, Version 23.0. Any *p* < 0.05 was regarded as statistically significant. Descriptive statistics of all outcomes were analyzed to identify the demographic distribution of response. One-way ANOVA and *t*-test were performed for comparing between groups. Pearson Correlation test was used to test for association between two variables. Univariate analysis was first conducted to identify the association between confidence and knowledge, attitude and exposure. Variables with *p* < 0.2 were further selected for the final model. Multivariate linear regression analysis was conducted to identify the associated factors controlling with the gender and age.

## 3. Results

### 3.1. Socio-Demographics

In total, 303 questionnaires were collected with a 65% of response rate which was advertised on social media through convenient sampling. Participants are invited to share the survey link to another respondent upon completion. The majority of them were female (62.7%) and had no religious belief (61.7%). Distribution of medical students between pre-clinical years (40.9%) and clinical year (59.1%) are similar. Demographic factors remained insignificant for the association with the confidence (Table 1). To facilitate the investigation, gender and age were selected into the final multivariate analysis for adjustment as a confounder with references to various abroad studies on the association [39,41].

### 3.2. Confidence to Provide Palliative Care

The average score of confidence to provide palliative care was 26.5 (SD: 7.5, 95% C.I.: 26.0–27.4) out of 50 with the range of 10–44. Overall, 59.4% of respondents were “non-confident” while 40.6% were “confident” towards providing palliative care. There was no statistically significant difference among all the demographics variables.

Primarily, high confidence of medical students in providing palliative care is significantly positively associated with higher level of knowledge of palliative care (*p* = 0.026) and more positive attitude towards palliative care provision (*p* = 0.014). Having more clinical exposure is preliminary significantly positively associated with confidence in palliative care provision (*p* = 0.186), whereas the exposure to death of family/friends has a negative association with high confidence in providing palliative care (*p* = 0.041). Knowledge, attitude, clinical exposure, and exposure to death of family/friends were selected to the final multivariate model for a further data analysis.

In the gender-age adjusted multiple linear regression, knowledge (*p* = 0.011, β = 0.407, 95% C.I.: 0.096 to 0.717) and attitude (*p* = 0.003, β = 2.581, 95% C.I.: 0.884 to 4.279) showed a positive association with confidence. On the contrary, the exposure to death of family/friends (*p* = 0.024, β = −1.996, 95% C.I.: −3.725 to −0.267) demonstrates a negative association with the confidence (Table 2).

## 4. Discussion

Although Hong Kong has a relatively advanced and responsive health system, the palliative care ranking (22nd) in terms of Quality of Death Index [26] is not as satisfactory. Similar to other countries, such as South Korea (18th), they also show a resonance to the mentioned trend. However, there was a lack of research on the inconsistency between the overall medical system and the respective palliative care services. Thus, this study was conducted to begin an investigation for the phenomenon in Hong Kong primarily, as well as provide international insights for other places to examine the reasons within their local context. From this study, 59.4% of responding students indicated being “non-confident” in providing palliative care. The results are consistent with most abroad studies, such as the Netherlands (59.6%) [24], Germany (68.5%) [30], Thailand (66.7%) [28], etc. which also show a tilted proportion towards “non-confidence” at around 60% to 70%. It indicates a resonance between countries in the relatively low estimation of students’ personal competency in upcoming palliative work. In the Netherlands (8th) and Germany (7th), which have a high ranking on the Quality of Death Index (EIA) [26], students’ confidence might potentially be affected by the expectation of the local community in the provision of high-quality medical services [4]. Meanwhile, in Thailand (44th), nurses are usually known as the care-coordinators in palliative care units instead of doctors [42]. Further, in addition to the limited education on palliative care during schooling, medical undergraduates might not recognize their roles and abilities to provide palliative care services for patients due to the construct set beforehand. They might tend to rely on nurses for the job to handle related clinical matters which leads to a low confidence to provide the palliative care [42].

Confidence could be influenced by multiple factors [3]. Despite the consistently low confidence level, most abroad studies were conducted to mainly emphasize the necessity of implementing palliative care courses into their respective medical curriculum in view of the lack of palliative education. Unlike other studies, the current Hong Kong medical curriculum has already been integrating palliative care into continuous ethical training and related courses (e.g., lectures, facilitated discussion groups, case studies, clinical exposure etc.) [12,13]. Compared to South Korea (18th) with a similar ranking as Hong Kong (22nd), the “non-confidence” rate researched among undergraduates is 14.4% [34]. The difference in results could be influenced by the rich resources implemented by the government, including a 60-h standard hospice and palliative care education program for all related disciplines in 2005 [43], and the 2.3 billion USD subsidization on palliative care units, through which, palliative care units increased from 15 (2005) to 44 (2011). With more human resources, medical students do not have to worry about an extreme upcoming workload to handle the demands of palliative care [43]. In addition, with a higher awareness and more education on palliative care, undergraduates could also have fewer negative perceptions over palliative care, and thus a higher confidence over providing palliative care in their future career [12]. Regardless of the similar setting, there could be other underlying factors influencing the results generated from the study in Hong Kong, for instance, to explain how more years of study does not necessarily mean a higher confidence (Appendix A). More research (e.g., qualitative study) is required to identify and look deeper into the reason for the phenomenon among local medical undergraduates in Hong Kong setting specifically.

### 4.1. Knowledge

In the study, knowledge is significantly positively associated to the confidence of students in providing palliative care. Such results are also consistent with abroad studies, where places like the Netherlands [27], Germany [30], the United Kingdom [44] and South Korea [45] have conducted research proving a significant association between knowledge and confidence. When students obtain more comprehensive knowledge and a better understanding of different aspects of palliative care (e.g., philosophy), it allows them to have a higher preparedness and hence, confidence, to begin their clinical work in the future [3,12]. Especially in the local medical curriculum combined with diverse formats of educations (e.g., clinical exposure), it helps student to have a better foundation on the knowledge and skills required to take care of palliative patients, thus giving them a higher confidence to stay competent in their future provision of clinical services [12,28,30]. In addition, under the teaching of professors and medical workers, misunderstanding and worries of students could be overcome with clear explanations by teachers and higher exposure [12]. Thus, they will have less concerns and higher confidence as a result. To understand more on the association, the knowledge on palliative care should be further assessed with a validated local/national tool targeting medical undergraduates in the future as the index might not apply to the local perspective, nor the study population.

### 4.2. Attitude

Attitude is significantly positively associated with the confidence in our study population. As a higher attitude scoring refers to more positive beliefs, and less perceived barriers and misunderstanding, which could possibly lead to a higher confidence among students in providing palliative care in the future. For instance, students with more fear towards death could result in a lower confidence as they might be afraid to witnessing the pass away of patients; or students who believe palliative care could be beneficial to their own growth and development might tend to have a higher confidence. In Taiwan, which was ranked 6th on the Quality of Death index [26], their studies also stated the beliefs of physicians are strongly predictive in their respective confidence to provide palliative care [12]. In Netherlands (8th) and Australia (2nd), it was researched that medical curriculum could improve the quality of care in terms of fostering positive beliefs and attitude [46,47]. Since medical curriculum are usually embedded with real-life clinical participation and thorough teaching, it could educate students with a better handling of complex palliative situations and amend their negative beliefs with appropriate means responsively [12]. Apart from the mentioned explanations, some articles also stated that a higher confidence could relate with more positive beliefs for clinical workers in providing care services [12,46]. Thus, it does require more research to be done to identify the directions of relationship between the variables and to understand whether they are mutually interactive.

### 4.3. Exposure

On the other hand, the death of the family/friends is significantly negatively associated with the confidence in our study. It could potentially indicate that when students faced the death of their loved one, their confidence to provide palliative care in the future could reduce. As physicians-to-be, medical undergraduates are more likely to face deaths at their clinical working setting than other populations. Thus, witnessing the death of patients could trigger and evoke their traumatic memories and sorrow feelings about their past loved ones. Palliative care varies from curative care, which is a care service to provide a good death for patients who are suffering from life-threatening diseases [1]. The clinical structural setting could therefore expose palliative care physicians to even higher opportunities to handle the deaths of patients. In order to maintain their emotions, professionality and quality of care, they might not hope to be personally affected by palliative cases, which gives them a lower confidence to provide palliative care [10]. However, the result among our study population is inconsistent with other studies in two aspects: (1) association and (2) the direction of the association. Another study conducted in 2007 in Hong Kong states that experience is not significantly associated with the preparedness of students on end-of-life care. The potential reason stated was the inadequate education [4]. Yet, during these years, the medical curriculum was continuously refining with evaluations and assessments, hence causing a change in association [3,17,33]. While in Taiwan, research shows that experience is positively associated with the willingness to provide palliative care among working doctors [12]. The variety in the direction of the results could be due to the difference in the target population of respective research. The mentioned research recruited working physicians as their respondents while this study focuses on medical undergraduates as the population. For undergraduates who still lack experience and maturity to tackle emotions compared to working doctors, they are more likely to build an emotional barrier in handling deaths of palliative patients due to sorrow. Additionally, clinical exposure was also investigated in the research, but it does not have a significant association with the confidence in the study. It might be due to the extent of influence of family/friends than patients to local students. Asian culture emphasizes filial piety and affection, which might also affect students’ confidence to provide care and explains the difference in the significance of associations [48,49]. From the open-ended questions collected from the questionnaire, junior medical students also reflect a lack of actual clinical experience to practice their palliative care knowledge which could influence the significance of association. Therefore, a more updated and in-depth research would be required to understand the reason behind this.

### 4.4. Limitations

With an objective view of the study, there are some limitations. As a cross-sectional study, it could not determine any causal relationship between confidence and the factors. Thus, we suggest conducting a longitudinal study to follow up with the respondents to better understand the individual relationships of factors and the directions of association. Moreover, self-administered questionnaires might not be well-understood by medical respondents in different medical years. Thus, pilot tests were conducted which allows researchers to make minor adjustments in phrases with less jargons to improve clarity. Due to COVID-19, enquiries were actively responded to make up for the unavailability of face-to-face interaction. The limited sample size would also be a limitation for sub-group analysis as the pandemic delayed the efficiency of data collection. Future research with a larger sample size with probability sampling can improve the generalizability of the sample and results, as well as help in understanding more information on the association (e.g., influence of medical training years).

## 5. Conclusions

The study demonstrated that the confidence in providing palliative care on average could be enhanced among medical students. Knowledge, attitude and exposure to death of family/friends are researched to be associated with the confidence on providing palliative care. The results resonate to abroad studies, with statistics showing that the local medical curriculum provides a comprehensive education for the medical students in terms of a refined academic understanding and fostered positive beliefs. The study could also provide insights to improve the overall confidence of medical undergraduates responsively, and ultimately refining the availability and sustainability of palliative care services in Hong Kong and around the globe. More in-depth research is therefore encouraged to investigate the research area more comprehensively.

## 6. Patents

This section is not mandatory but may be added if there are patents resulting from the work reported in this manuscript.

## Figures and Tables

**Table 1 ijerph-18-08071-t001:** Association between demographics and confidence towards providing palliative care among medical students.

Socio-Demographics	No. of Respondents N (%)	Confidence Score (Mean ± SD)	*p*-Value
**Sex**			0.517
Female	190 (62.7)	26.3 ± 7.59	
Male	113 (37.3)	26.9 ± 7.43	
**Age**			0.193
</=20	130 (42.9)	27.2 ± 7.82	
>20	173 (57.1)	26.0 ± 7.28	
**Years of Study**			0.388
Years 1–3 (Pre-clinical Years)	124 (40.9)	27.0 ± 8.24	
Years 4–6 (Clinical Years)	179 (59.1)	26.2 ± 7.00	
**Religion Beliefs**			0.607
Without religion beliefs	187 (61.7)	26.5 ± 7.59	
With religion beliefs	116 (38.3)	25.6 ± 3.95	
**Occupational Status of Breadwinner**			0.338
White Collar	247 (81.5)	26.4 ± 7.69	
Blue Collar	43 (14.2)	26.3 ± 6.85	
Housewives/Retired/Unemployed	13 (4.3)	29.2 ± 6.41	
**Institution**			0.282
CUHK	196 (64.7)	26.2 ± 7.59	
HKU	107 (35.3)	27.1 ± 7.40	
**Total**	303 (100.0)	26.5 ± 7.53	

**Table 2 ijerph-18-08071-t002:** Factors associated with the confidence towards providing palliative care among medical students.

Variable	Total	Confidence Score			
	Mean ± SD	Correlation	*p*-Value ^~^	Coef (95% C.I.) ^#^	*p*-Value ^#^
Knowledge (20 items)	6.49 ± 2.84	0.128	**0.026 ***	0.407 (0.096, 0.717)	**0.011 ***
Attitude (10 items)	3.82 ± 0.51	0.141	**0.014 ***	2.581 (0.884, 4.279)	**0.003 ***
Clinical Exposure (no. of times)	0.79 ± 2.29	0.076	0.186	0.232 (−0.139, 0.603)	0.221
	N (%)	Mean ± SD	*p*-value ^		
Family/Friends Exposure (Y/N)			**0.047 ***		**0.024 ***
No	111 (36.6)	27.7 ± 7.96		ref	
Yes	192 (63.4)	25.8 ± 7.20		−1.996 (−3.725, −0.267)	

* *p*-value < 0.05. ~ Pearson Correlation test was used. ^ *t*-test was used. ^#^ Gender-age adjusted multiple linear regression was performed.

## Data Availability

Not applicable.

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
