# Peer review of "Assessing Medical Students’ Confidence towards Provision of Palliative Care: A Cross-Sectional Study"

_ijerph, 2021, doi:10.3390/ijerph18158071_

Round 1
Reviewer 1 Report
The revision was minor by adding two references and a few short paragraphs. It is true that since all the data were collected and it is hard to collect new data.
Reviewer 2 Report
The revisions have been incluided and the paper has been improve.
This manuscript is a resubmission of an earlier submission. The following is a list of the peer review reports and author responses from that submission.
Round 1
Reviewer 1 Report
It is certainly an important area of investigation as medical students were blinded by slogans of "Doctors Cure". In reality, most chronic or advanced illnesses have no cure, doctors can only provide palliative care and end of life care. Providing appropriate education, training, exposure, simulation programs, attachment, etc. for medical students are certainly important as more population are facing rapid aging and more people are living with advanced diseases that requires palliative care. With severe unconscious bias for "cure" in the medical education, there is a great need for critically review of the current status of self assessments on their own confidence in palliative care. Yet, the concepts of "self-assessed competency" may be a better framework than the current idea of "confidence".
There are two universities offering medical education, students from both of these units were recruited into this cross sectional study. However, the description of the questions were confusing and unclear, for example, career choice (white collar, blue collar, professional...) sound really odd as the respondents were medical students. It was later described that the "career choice" was about the occupation of the breadwinner of the household. How is breadwinner's job conceptually linked to palliative care self confidence among the medical students?
Confidence was being measured by "correct" answers and linked to selected knowledge, attitudes, clinical and personal exposure to palliative care. Then, respondents were classified into confident and non-confident. Then, relationships among these variables were developed. The findings are un-interesting and no new knowledge being generated.
Unfortunately, there is a lack of strong and original conceptual or theoretical framework in this investigation, making it a simple opinion survey on attitudes on palliative care among a few hundred of medical students in Hong Kong. Findings are similar to survey of other places, and there is not much new knowledge being generated by this study.
There are also language problems. The term "Non-Confident" should be "No- OR Not-Confident". Non-Confident usually means something quite different. "Abroad studies" may also be worded as previous studies or international studies?
Reviewer 2 Report
I have read with interest this paper. I believe that the paper is interesting. However, I have some concerns that are reported herein.
Introduction: I recommend to include the term “compassion”. Authors can consult the paper: doi:10.3390/ijerph17155425
Variables: Authors should define all outcomes, exposures, predictors, potential confounders, and effect modifiers.
Bias and study size: authors should describe any efforts to address potential sources of bias and to explain how the study size was arrived at.
Stadistical methods: authors should explain how missing data were addressed.
Limitations: authos should discuss limitations of the study, taking into account sources of potential bias or imprecision. Discuss both direction and magnitude of any potential bias